# Observations of the thermodynamic and kinematic state of the atmospheric boundary layer over the San Luis Valley, CO using the CopterSonde 2 remotely piloted aircraft system in support of the LAPSE-RATE field campaign

Elizabeth A. Pillar-Little[1,2], Brian R. Greene[1,2,3], Francesca M. Lappin[1,2], Tyler M. Bell[1,2,4,5], Antonio R. Segales[2,3,6], Gustavo Britto Hupsel de Azevedo[2,6], William Doyle[2], Sai Teja Kanneganti[2,7], Daniel D. Tripp[4,5], and Phillip B. Chilson[1,2,3]

[1]School of Meteorology, University of Oklahoma, Norman, OK 73072, USA
[2]Center for Autonomous Sensing and Sampling, University of Oklahoma, Norman, OK 73072, USA
[3]Advanced Radar Research Center, University of Oklahoma, Norman, OK 73019, USA
[4]Cooperative Institute for Mesoscale Meteorological Studies, University of Oklahoma, Norman, OK 73072, USA
[5]NOAA/OAR National Severe Storms Laboratory, Norman, OK 73072, USA
[6]School of Electrical and Computer Engineering, University of Oklahoma, Norman, OK 73019, USA
[7]School of Computer Science, University of Oklahoma, Norman, OK 73019, USA

**Correspondence:** Elizabeth A. Pillar-Little (epillarlittle@ou.edu)

**Abstract.** In July 2018, the University of Oklahoma deployed three CopterSonde remotely piloted aircraft systems (RPAS) to take measurements of the evolving thermodynamic and kinematic state of the atmospheric boundary layer (ABL) over complex terrain in the San Luis Valley, Colorado. A total of 180 flights were completed over five days, with teams operating simultaneously at two different sites in the northern half of the valley. Two days of operations focused on convection initiation studies, one day focused on ABL diurnal transition studies, one day focused on internal comparison flights, and the last day of operations focused on cold air drainage flows. The data from these coordinated flights provides insight into the horizontal heterogeneity of the atmospheric state over complex terrain. This dataset, along with others collected by other universities and institutions as a part of the LAPSE-RATE campaign, have been submitted to Zenodo (Greene et al., 2020) for free and open access (DOI:10.5281/zenodo.3737087).

## 1 Introduction

Researchers from the University of Oklahoma (OU) joined colleagues from across the world to take part in the Lower Atmospheric Profiling Studies at Elevation – a Remotely-piloted Aircraft Team Experiment (LAPSE-RATE) campaign during 13-19 July 2018 in the San Luis Valley of Colorado. This campaign brought teams together from several universities with researchers from government laboratories such as the National Center for Atmospheric Research (NCAR), National Severe Storms Labo-

ratory (NSSL) and National Oceanic and Atmospheric Administration (NOAA) to conduct targeted observations to examine five scientific objectives: 1) valley drainage flows, 2) convection initiation, 3) aerosol properties, 4) turbulence profiling, and 5) morning boundary layer transitions with both remote (LIDARs, AERI, radiometer) and in situ (radiosondes, Mobile Mesonet) sensors (Bell et al., 2020b; de Boer et al., 2020c) as well as remotely piloted aircraft systems (RPAS, also commonly referred

to as uncrewed aircraft systems, UAS). Additionally, the colocation of so many teams with diverse systems provided a unique opportunity to undertake an intensive comparison of the sensing capabilities of the aerial systems being utilized as a part of the campaign (de Boer et al., 2020b; Barbieri et al., 2019). It also provided an opportunity to assess the accuracy of weather forecasts that were provided by NCAR to the team as a part of the campaign (Glasheen et al., 2020; Pinto et al., 2020).

The use of RPAS in the atmospheric sciences has increased significantly with the explosive growth of technologies that are
both economical and more user-friendly than previous generations of uncrewed systems (Elston et al., 2015; Brosy et al., 2017; Koch et al., 2018; Lee et al., 2017, 2018; Chilson et al., 2019). Research utilizing RPAS for weather applications at OU has been ongoing since 2009, when scientists in the School of Meteorology began exploring ways of utilizing RPAS to take highly resolved profiles of atmospheric state variables (Bonin et al., 2013, 2012), turbulence (Wainwright et al., 2015; Bonin et al., 2015), and ozone (Zielke, 2011) among other phenomena to better understand the evolution and structure of the atmospheric
boundary layer (ABL). In 2016, these studies expanded to encompass sensor placement and measurement optimization (Greene et al., 2018), system design and evaluation (Segales et al., 2020a), and sensor integration (Greene et al., 2019) due in large part to the CLOUD-MAP project which facilitated the development of RPAS for the explicit purpose of conducting atmospheric measurements (Jacob et al., 2018). The capabilities of these weather-sensing RPAS have been demonstrated in a variety of collaborative field campaigns (de Boer et al., 2019; Jacob et al., 2018; Koch et al., 2018; Kral et al., 2020), calibration and
validation experiments (Barbieri et al., 2019), and careful comparison against other remote sensing networks (Bell et al., 2020a).

OU deployed three CopterSonde quadcopter RPAS (Segales et al., 2020a) as a part of the LAPSE-RATE campaign. This system successfully captured vertical profiles of these variables up to 914 m above ground level (AGL) during this week-long mission through direct (pressure, temperature, and humidity) and indirect (wind speed and wind direction) measurements (see
Tables 1 and 2 for technical specifications). These platforms were deployed to three different sites over the course of the field campaign, typically measuring at two sites simultaneously. This approach resulted in the OU team completing 180 flights that produced quality observational data supporting three of the five campaign objectives (boundary layer transitions, drainage flows, and convection initiation).

This manuscript will focus on the data collected by OU in support of the scientific objectives of the campaign, including
the platforms used, the data acquisition methods, and the processing and archival process. Section 2 will briefly describe the CopterSonde RPAS as well as the operational strategy for this field campaign. Section 3 will discuss the data logging, processing, and quality control procedures, while Section 4 will highlight some examples of data products available within this dataset. Section 5 will outline the format, location, and associated metadata of the publicly available dataset. Finally, Section 6 will provide concluding remarks as future outlooks regarding the future applications of the dataset. For details regarding OU's
contribution to the sensor intercomparison efforts, readers are directed to Barbieri et al. (2019). More information regarding

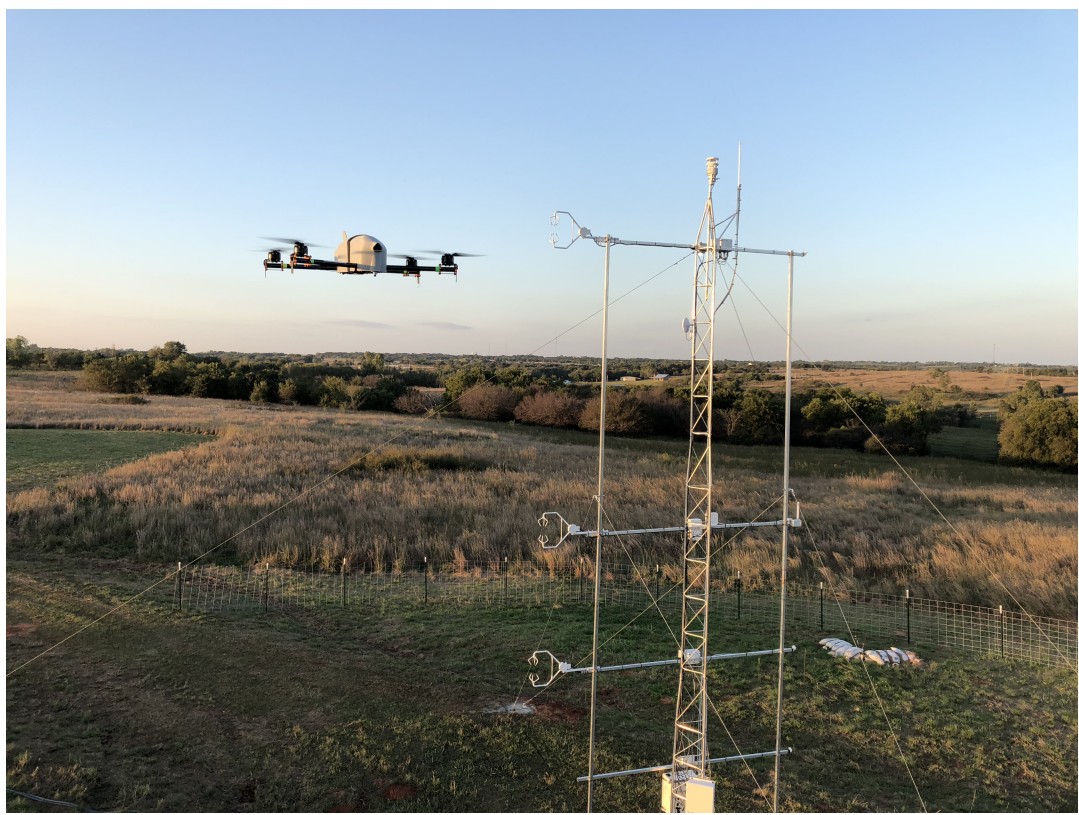

**Figure 1.** Photograph of the OU CopterSonde RPAS (Norman, OK, USA) in flight next to an experimental 10 m flux tower.

the overall campaign such as information surrounding the synoptic conditions, community engagement efforts, and campaign objectives can be found in de Boer et al. (2020a) and de Boer et al. (2020b). Finally, material outlining the contribution of the other participating institutions can be found in this special issue (Bailey et al., 2020; Bell et al., 2020b; Brus et al., 2020; de Boer et al., 2020c; Pinto et al., 2020).

## 2   Description of the CopterSonde RPAS and Flight Strategy

### 2.1   CopterSonde RPAS

The RPAS utilized for this field campaign was the CopterSonde 2 (hereinafter, CopterSonde) (Figure 1; see Segales et al., 2020a), which was designed and manufactured by the Center for Autonomous Sensing and Sampling (CASS) at OU. The CopterSonde is a rotary-wing platform that is based on a modified version of the Lynxmotion HQuad500 wide-X type quad-copter with fixed-pitch rotors that has been optimized for vertical profiling operations. The CopterSonde's technical specifications can be found in Table 1. Additional information regarding the design and development of the CopterSonde series is described in Segales et al. (2020a). The CopterSonde RPAS has been proven to be capable of collecting thermodynamic and

**Table 1.** Technical Specifications of the OU CopterSonde

| | |
|---|---|
| Frame size | 500 mm |
| All-up weight | 2.25 – 2.36 kg |
| Maximum speed | 26.4 m s$^{-1}$ |
| Maximum ascent rate | 12.2 m s$^{-1}$ |
| Maximum descent rate | 6.5 m s$^{-1}$ |
| Maximum altitude above ground[a] | 1800 m |
| Maximum altitude above sea level | 3050 m |
| Maximum wind speed tolerance | 22 m s$^{-1}$ |
| Flight endurance[a] | 18.5 min |
| Operating temperatures[b] | $-20\,°C – 40\,°C$ |
| Measured Thermodynamic Variables | Temperature, Pressure, Relative Humidity |
| Derived Kinematic Variables | Wind Speed, Wind Direction |

[a] under favorable weather conditions with low winds.

[b] tested temperatures, the range can be larger than stated.

kinematic profiles of the atmosphere in a variety of environments from summer in the Southern Great Plains in pre-convective environments (Koch et al., 2018) to polar winter conditions in the Arctic (Kral et al., 2020). For the LAPSE-RATE deploy-
65 ment, the CopterSonde was piloted under its typical operating parameters except its standard 11" x 5.5" T-style carbon fiber propellers were swapped for a 12" x 5" model to increase the maximum thrust of the vehicle required to overcome the high density altitude in south central Colorado resulting from warm temperatures and high altitudes above mean sea level (MSL). Because the thermodynamic sensors are mounted inside a fan-aspirated L-duct on the front end of the aircraft, results from Greene et al. (2019) indicate that changing the propellers should not affect the quality of thermodynamic observations. All
70 the CopterSonde's flights during the campaign were done in a semi-autonomous mode, meaning that the platform flew a pre-programmed mission and was only manually controlled by the operator during landing. Commands were sent to the RPAS over a telemetry link and data collected were streamed back to the ground station where they were displayed on a customized interface that allowed for real-time monitoring. The ground station and RPAS communicated via a 900 MHz Radio (RFD 900+, RFD Design) that has a range of 40 km.
The CopterSonde is outfitted with sensors that enable it to measure atmospheric state variables as it ascends along its flight path (see Table 2 for details). The wind speed and wind direction were calculated at 10 Hz using the Wind Vane Mode algorithm described in Segales et al. (2020a) which utilizes the roll, pitch, and yaw angles measured with the inertial measurement unit (IMU) on-board the RPAS's autopilot system, the Pixhawk CubeBlack. The pressure was measured at 10 Hz with a MS561 capacitive pressure sensor inside the Pixhawk CubeBlack, which is also utilized for altitude. Atmospheric temperature was
measured at 20 Hz with a fast response bead thermistor (International Met Systems). Relative humidity was measured at 10 Hz using the HYT 271 capacitive humidity sensor (Innovative Sensor Technologies). As will be discussed later, data from the

**Table 2.** Description of the thermodynamic and kinematic parameters available in each CopterSonde data file. Also includes information on the method each parameter was measured along with their relative accuracies based on Bell et al. (2020a) compared to Vaisala RS92-SGP radiosondes.

| Parameter | Method | System Accuracy |
|---|---|---|
| Temperature | 3 iMet-XF Bead Thermistors | $\pm 0.5°$C |
| Relative Humidity | 3 IST HYT-271 Capacitive Hygrometers<br>Converted to dewpoint temperature then recalculated<br>using (relatively) fast-response iMet-XF bead thermistors | $\pm 2\%$ |
| Dewpoint Temperature | Converted from temperature and relative humidity<br>measured by the 3 IST HYT-271 sensors | $\pm 0.5°$C |
| Pressure | TE MS5611 Barometric Pressure Sensor (inside Pixhawk 2.1 Autopilot) | $\pm 1.5$ hPa |
| Wind Speed | Linear regression of tilt angles<br>compared to reference | $\pm 0.6$ m s$^{-1}$ |
| Wind Direction | Linear regression of tilt angles<br>compared to reference | $\pm 4°$ |

different sensors were interpolated or downsampled so that all observations have a common time vector. The temperature and humidity sensors were enclosed in a custom sensor scoop that was 3D printed out of polylactic acid (PLA). The sensors were located inside the tubular portion of an L-shaped duct and were mounted in an inverted V configuration. At the base of the duct
was a smart fan that was programmed to aspirate the sensors at a rate of 12 m s$^{-1}$ and was toggled on during ascent at a height of 1.85 m AGL and off during descent at a height of 1.45 m AGL to prevent dust and debris from being pulled into the scoop. Each scoop was distinguished utilizing an identification code and calibrated prior to the field campaign using the procedure outlined in Greene et al. (2019). Further information regarding considerations for sensor placement, aspiration, and shielding on the CopterSonde can be found in Greene et al. (2018, 2019). More on the data quality and statistical performances will be
discussed in Section 3.

## 2.2   Flight Strategies

The LAPSE-RATE campaign featured five unique scientific objectives: morning atmospheric boundary layer transitions, aerosol properties, valley drainage flows, deep convection initiation, and atmospheric turbulence profiling. Each evening, the individual LAPSE-RATE teams would gather for a weather briefing and discuss which objectives would be the most advanta-
geous to target based on forecasts prepared by NCAR scientists. Teams would then distribute themselves across the San Luis Valley to best sample the phenomena of interest. For more detailed information about the overall campaign goals, science objectives, synoptic and mesoscale conditions driving the selection of objectives, please see the LAPSE-RATE overview article in this special issue (de Boer et al., 2020b).

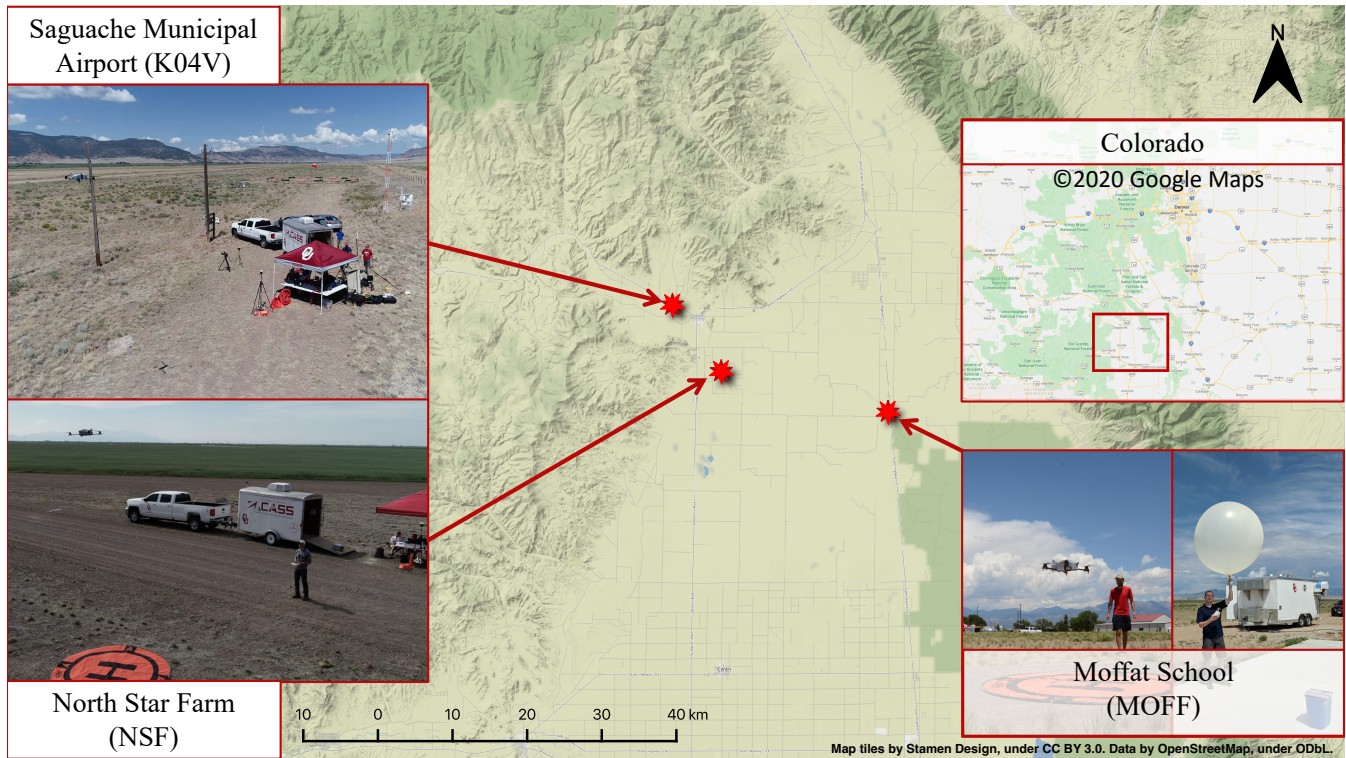

**Figure 2.** Map of the San Luis Valley in south-central Colorado (data ©2020 OpenStreetMap), with CASS deployment locations denoted by the red stars. Inset images beginning from the top left and moving counterclockwise: Saguache Municipal Airport (K04V, photo credit William Doyle), North Star Farm (NSF, photo credit William Doyle), Moffat School (MOFF, photo credit Tyler Bell), and a map of the state of Colorado with the San Luis Valley outlined in red (courtesy Google Maps, ©2020).

The CopterSonde was deployed at three different locations across the San Luis Valley in support of LAPSE-RATE scientific
objectives. These locations were Saguache Municipal Airport (K04V), Moffat Consolidated School (MOFF), and the southern edge of North Star Farms (NSF). Latitude and longitudes of these deployment locations as well as their altitude above mean sea level (m MSL) are summarized in Table 3. A visual representation of these locations with respect to the layout of the valley is provided in Figure 2, with the inset focusing on the layout of the multiple assets operating at MOFF. The MOFF and NSF sites were both located in the central portion of the valley whereas K04V was situated in a narrowing section of the valley
with steeper valley walls to the northwest of NSF. K04V and MOFF were dominated by scrublands while NSF was primarily comprised of irrigated cropland.

OU deployed a team daily to MOFF regardless of objective due to the colocation of the Collaborative Lower Atmosphere Mobile Profiling Station (CLAMPS) at the site. Not only did this create a local supersite in the central region of the valley, it facilitated the intercomparison of the RPAS with more conventional instrumentation such as radiosondes, atmospheric emitted
radiance interferometer (AERI), microwave radiometer (MWR), and Doppler LIDAR (Bell et al., 2020a; Wagner et al., 2019).

**Table 3.** List of sites during LAPSE-RATE where CASS operated.

| Site (Abbreviation) | Latitude (°N) | Longitude (°E) | Altitude (m MSL) |
|---|---|---|---|
| Saguache Municipal Airfield (K04V) | 38.099156 | -106.171331 | 2385.2 |
| Moffat School (MOFF) | 37.997587 | -105.911795 | 2305.6 |
| North Star Farm (NSF) | 38.036093 | -106.112658 | 2330.8 |

**Table 4.** Summary of flights for each day broken down by aircraft and location. Local time during the campaign was Mountain Daylight Time (UTC−6).

| Date | Time Start | Time Stop | Science Objective | Site 1 | Aircraft Number | Flight Count | Site 2 | Aircraft Number | Flight Count | Daily Total |
|---|---|---|---|---|---|---|---|---|---|---|
| 14 July 2018 | 1430 UTC | 2130 UTC | CASS Tests | MOFF | OU944 | 4 | – | – | – | 4 |
| 15 July 2018 | 1330 UTC | 1945 UTC | CI | MOFF | OU944 OU955 | 2 18 | K04V | OU946 | 14 | 34 |
| 16 July 2018 | 1330 UTC | 2115 UTC | CI | MOFF | OU955 | 26 | K04V | OU946 | 21 | 47 |
| 17 July 2018 | 1330 UTC | 2130 UTC | CASS Tests | MOFF | OU944 | 5 | MOFF | OU946 | 9 | 14 |
| 18 July 2018 | 1230 UTC | 1945 UTC | ABL Transition | MOFF | OU944 | 14 | K04V | OU946 | 21 | 35 |
| 19 July 2018 | 1115 UTC | 1700 UTC | Drainage | MOFF | OU944 | 24 | NSF | OU946 | 22 | 46 |
| | | | | **MOFF Total** | | 93 | **K04V & NSF Total** | | 87 | **180** |

A second OU team was deployed to one of three locations depending on the daily objective. For the convection initiation (CI) and boundary layer transition (BLT) study days, the second team set up at K04V. For the cold air drainage study, the team deployed to NSF. Across all three sites, 180 successful flights were completed in support of LAPSE-RATE. Details regarding the number of flights per day, daily science objectives, start and stop times, and specific vehicles used are summarized in Table 3.

All flights completed by OU as a part of LAPSE-RATE were conducted either under Federal Aviation Authority (FAA) Part 107 regulations or under the Oklahoma State University's FAA Certificate of Authority (COA) 2018-WSA-1542 effective from 13 - 22 July 2018. This COA permitted daytime operations of small RPAS weighting less than 25 kg (55 lbs.) at speeds less than 45 m s$^{-1}$ (87 kts) in Class E and G airspace below 914 m AGL and not exceeding 3,657 m MSL in in the vicinity of Alamosa County, CO under the jurisdiction of the Denver Air Route Traffic Control Center (ARTCC). Typical COA provisions

for airworthiness, operations, safety protocols, Notice to Airmen (NOTAMs), reporting, and registration were applied; however, special provisions were necessary for the coordination and deconfliction of the myriad of teams participating in flight operations as part of LAPSE-RATE. Instead of requiring each individual team to submit NOTAMs, discussions between LAPSE-RATE participants, the FAA, and Denver ARTCC lead to the definition of a common area of operations that could cover the entirety of the planned LAPSE-RATE observations. Additional deconfliction was also necessary with nearby airports, Military Training Routes (MTRs), or other restricted airspaces such as the Great Sand Dunes National Park. Within this area, two NOTAM subareas were defined so that the most appropriate areas could be activated with the necessary 24-hr notice based on the next day's scientific objectives. Emergency procedures for lost links, radio communications, and other potential anomalies also had special provisions due to the number of teams operating in proximity to each other. In addition to the COA provisions, each OU operations area was overseen by a licensed private pilot who assisted with overseeing the airspace and deconflicting RPAS operations from general aviation traffic.

CopterSonde missions were programmed to fly a vertical ascent from the surface to 914 m AGL, utilizing the platforms wind vane mode to continuously orient itself into the wind. This permitted the RPAS to sample the vertical structure of pressure, temperature, humidity, wind speed, and wind direction in a controlled and repeatable manner that minimized influences from the platform itself (Segales et al., 2020a). These flights will be referred to as profiles in subsequent text and tables. These profiles consisted of an automatic takeoff, vertical ascent at a rate of 3.5 m s$^{-1}$, loiter for 10 s at the apex of the ascent, and controlled decent to 10 m at a rate of 6 m s$^{-1}$. Once the platform completed its decent, it would be brought in for a landing manually. As will be discussed in Section 3, only the ascent portion of these vertical profiles are considered for analysis. We therefore chose to fly slower on the ascent to maximize the vertical resolution when accounting for thermodynamic sensor response times. Moreover, by descending more rapidly we are able to achieve a higher maximum profile altitude than we would otherwise with the same battery configuration on the CopterSonde.

Profiles were conducted on a 15- or 30-min cadence depending on the day's primary scientific objective and how rapidly the thermodynamic and kinematic parameters of the ABL were evolving. As the CopterSonde was collocated with CLAMPS at MOFF, RPAS profiles would often coincide with radiosonde launches. When this would occur, the RPAS launch would be held until the balloon cleared the airspace (about 60 s), and then would proceed as normal.

## 3  Data processing

The data collected by the CopterSondes were processed and quality controlled by CASS after the conclusion of the LAPSE-RATE campaign. The CopterSondes' Pixhawk autopilot system output and store a binary file on an on-board SD card during each flight, which includes logs of the aircraft's attitude angles, GPS positions, accelerations, and readings from the 3 temperature and 3 relative humidity sensors. In this format, the data are equivalent to the United States Department of Energy (DoE) Atmospheric Radiation Measurement (ARM) program's data archive "a0" level. Because the sensors log at different rates to the SD card, the binary files were converted to JavaScript Object Notation (JSON) format and relevant data parameters were interpolated/downsampled to a common 10 Hz time vector in comma-separated values (CSV) format (a1 level). The JSON file

format allows for the varying sampling rates for each data stream to coexist in the same file, whereas the conversion to CSV with a common time vector markedly simplifies reading and processing the data at this stage. More information about how the CopterSonde fuses sensor readings with autopilot features can be found in Segales et al. (2020a), and the autopilot code is freely available at Segales et al. (2020b).

After converting the raw binary flight log files to csv format, offsets for each temperature and relative humidity sensor were applied. These offsets were determined in the manner described by Greene et al. (2019) and Segales et al. (2020a), which involved isolating the CopterSondes' front L-duct sensor payloads in an environmentally-controlled chamber operated by the Oklahoma Mesonet with National Institutes of Standards and Technology (NIST) traceable sensors as references. Furthermore, at this stage, the CopterSonde attitude angles were averaged to estimate the horizontal wind speed and direction using linear regression coefficients determined by hovering next to Oklahoma Mesonet towers as a reference. This process is outlined in Greene (2018) and Segales et al. (2020a) (based on Neumann and Bartholmai, 2015; Palomaki et al., 2017).

Once the thermodynamic and kinematic calibrations were accounted for, the post-processing algorithm objectively determined the window of time for evaluation. This was chosen to be between 6 m AGL and the maximum point of the direct vertical profile (typically 914 m AGL during LAPSE-RATE) on the *ascending* portion only, averaged to 3-m bins as estimated by the Pixhawk autopilot's barometer and GPS sensors (Segales et al., 2020a). These criteria were chosen for several reasons, primarily to do with the relationships between ascent rate and sensor response time. Data averaging began at 6 m so as to avoid any possible contamination due to propeller wash interacting with the ground. The ascent portion is chosen because the flight patterns of the CopterSonde were chosen to maximize the achievable flight altitude and involves descending much more rapidly than ascending. Therefore, the descent portion suffers more from thermodynamic sensor response time issues. Furthermore, the physics behind the method of estimating horizontal wind speeds are not the same given the body forcings on a descending rotary-wing RPAS. Finally, the 3 m averaging interval was chosen under consideration of the average ascent rate (3.5 m s$^{-1}$) and an approximate time constant of the sensor payload of 2 s. This time constant is based upon experiments during the ISOBAR18 campaign with an older version of the CopterSonde and identical sensors (Kral et al., 2020; Greene et al., 2021, in preparation), where the aircraft was subjected to a series of quasi-step-function inputs between a sauna and the below-freezing environment of Hailuoto, Finland. The averaging interval of 3 m is therefore approximately double the vertical resolution as predicted by the response time and ascent rate, so further studies will be needed to elucidate the impacts of these decisions.

Because the CopterSondes were outfitted with 3 temperature and 3 RH sensors each, it was necessary to inspect each of their time-series outputs with respect to one another to determine potential outliers. Although an objective method of doing so is ideal, research into this is still ongoing and thus we chose to subjectively analyze each sensor individually. A given sensor was omitted from further consideration if it did not correlate with the other sensors and/or there was a large bias between them (greater than 0.5°C). All remaining sensor data were averaged together to yield a single measurement of temperature and relative humidity at each 3-m altitude interval. With an average ascent rate of 3.5 m s$^{-1}$ and a 10 Hz sampling rate, this corresponds to 8–9 samples per sensor per altitude averaging bin. These post-processed ascent profiles were then exported in

netCDF format (b1 level) that contain self-describing metadata including e.g., the specific aircraft and flight description. These file contents are described in Table 2.

In an effort to quantify the CopterSonde thermodynamic and kinematic observational biases relative to a ubiquitous standard, Bell et al. (2020a) compared vertical profile CopterSonde flights from LAPSE-RATE and in Oklahoma to collocated Vaisala RS92-SGP radiosondes. While unable to explicitly account for factors such as horizontal heterogeneity, the sample ranges in temperature, dewpoint temperature, and horizontal winds were large enough to determine baseline accuracies in each (Table 2). Namely, CopterSonde temperatures were within 0.5°C of the radiosondes in the aggregate, which is largely due in part to the considerations taken for temperature sensor placement on-board the CopterSonde (Greene et al., 2018, 2019). Additionally, a broad intercomparison effort during the LAPSE-RATE campaign (Barbieri et al., 2019) resulted in similar statistics when comparing the CopterSonde observations to a common mobile meteorological reference.

## 4 Examples of Flight Data

### 4.1 Convective Initiation – 15 July 2018

The environment on the morning of 15 July 2018 was rich in moisture with decreasing stability as the day progressed. This made for good conditions to target pre-convective measurements. The low wind shear and weak synoptic flow were conducive for isolated CI. This aided in spatially discerning precursors to CI. Two CASS teams were stationed at K04V and MOFF, approximately 18 km apart, with aircraft OU946 and OU955, respectively. Both teams began flying at 1400 UTC (0900 MDT). The team at K04V flew profiles every 30 minutes until 1830 UTC (1330 MDT). The team at MOFF had profiles every 15 minutes until 1945 UTC (1445 MDT). The cadence was reduced to 30 minutes once the ABL had become mixed at 1500 UTC (1000 MDT). At 1830 UTC (1330 MDT), there was convection in the vicinity and flight cadence was increased to 15 minutes at both sites. The maximum flight height for both teams was 914 m. Figures 3a and 3b show differences in how moisture and temperature evolve with time at each site. Figure 3a shows specific humidity increasing at K04V with time, meanwhile decreasing at MOFF (Figure 3b). Evaporation of additional moisture over MOFF slowed daytime heating. Although both sites had similar temperature fields, Figure 3a shows the temperature increasing faster with time. Since K04V was slightly drier, this led to faster destabilization and possibly preferential CI. Figure 3a also shows cooler temperatures below 300 m at 1800 UTC. At 100 m, K04V is 2 °C cooler than at MOFF. This may have been caused by outflow from a nearby storm. Even though the sites were not very far apart, there is a difference in the evolution of specific humidity. More cases would need to be analyzed to determine if this could have implicated the location of CI. Further analysis of the evolution of low-level buoyancy preceding this case is conducted in Lappin and Chilson (2021, in preparation). Previously, buoyancy has been used as a bulk stability parameter to determine storm severity (Zhang and Klein, 2010; Trier et al., 2014). This study looks into using buoyancy as a prognostic variable sensitive in time and space. It aims to discern local differences in ABL evolution in convective environments and further understanding of CI.

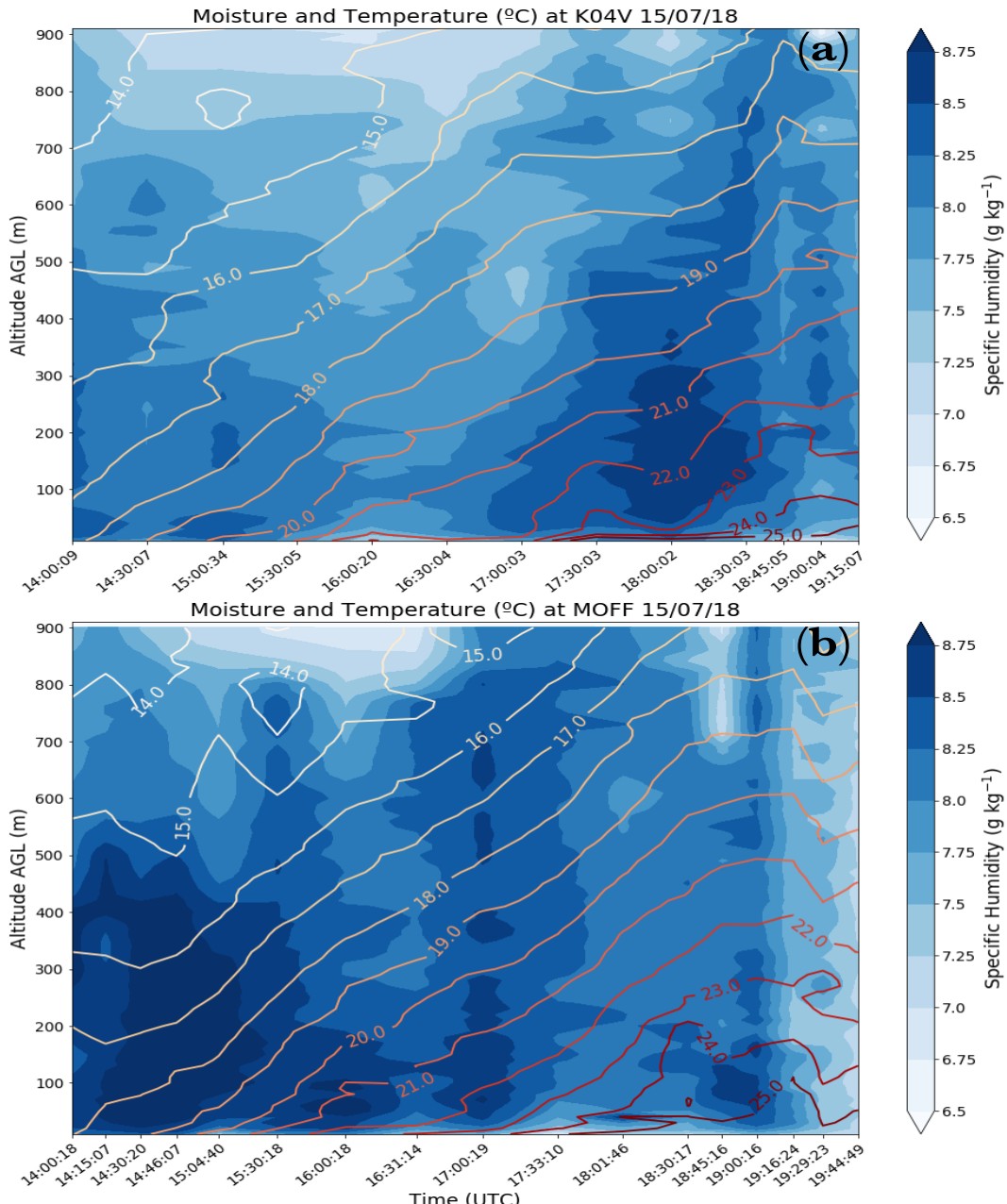

**Figure 3.** Time-height cross-section view of moisture and temperature measurements 15 July 2018. a) K04V site. b) MOFF site. This day experienced deep convection. Each time tick indicates a flight. Shaded field is specific humidity (g kg$^{-1}$). Contours are temperature ($^\circ$C). Values are interpolated through time at each level between successive flights.

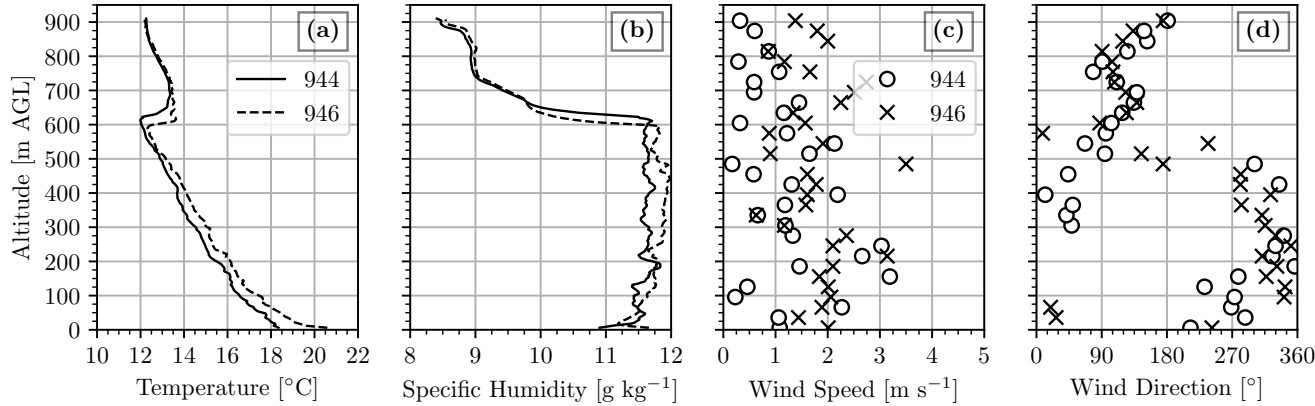

**Figure 4.** Comparisons of two CopterSonde flights launched simultaneously at 16:02:55 UTC at MOFF on 17 July 2018. Shown here are (a) temperature (°C), (b) specific humidity (g kg$^{-1}$), (c) horizontal wind speed (m s$^{-1}$), and (d) horizontal wind direction (degrees) versus altitude above ground level (AGL). Note that all data are included in (a) and (b), but data are subsampled at every eight points in (c) and (d) for clarity. In (a) and (b) ((c) and (d)), the solid and dashed black lines (open circles and Xs) represent the OU944 and OU946 aircraft, respectively. Wind speed and direction in general throughout the campaign were considerably variable in time and space.

## 4.2 CASS Test Flights – 17 July 2018

The LAPSE-RATE participants collectively decided for 17 July 2018 to be utilized for individual group research objectives, and so both mobile CASS teams decided to combine at the MOFF site for intercomparison flights between the CopterSondes and against CLAMPS. Between OU944 and OU946, 14 total vertical profile flights were conducted throughout the afternoon, 8 of which were simultaneous for direct comparisons across platforms (Table 4). Several of these flights were also accompanied by radiosonde launches directly before the vertical profiles, which were included along with the CLAMPS AERI and Doppler

LIDAR observations in the comparison study by Bell et al. (2020a). The LAPSE-RATE ground-based remote sensing data is outlined by Bell et al. (2020b). During these flights, the pair of CopterSondes flew about 10–20 m apart horizontally, and were also displaced about 50 m from the radiosonde launch site. An example of a direct comparison between OU944 and OU946 on the afternoon of 17 July (Figure 4) shows that both aircraft observed similar thermodynamic features with a well-mixed, dry adiabatic atmosphere up to around 600 m AGL, above which is notably drier and warmer than the ABL below. Although the

general profile shapes and inversion magnitudes are consistent across the platforms, a bias in temperature is apparent especially in the lowest 100 m. These discrepancies in the temperatures between the two identical platforms can likely be attributed to three main sources: 1) sunlight on an inadequately shielded sensor (discussed in Greene et al., 2019) at the correct relative angles of aircraft heading and sun zenith/azimuth; 2) natural variability in the atmosphere: the two aircraft were 10—20 m apart, so this is not entirely unreasonable for a convective boundary layer; and/or 3) systemic bias related to calibration of the

CopterSonde thermodynamic sensor package as a whole. While a combination of these three is the most likely explanation, we believe the spatial/temporal heterogeneity of the atmosphere during these observations should not be overlooked. For example,

3 s sonic anemometer temperatures from Bailey et al. (2020) reveal that during the 10 min timeframe during these concurrent CopterSonde profiles (albeit at a different site but featuring similar land cover properties), 2 m temperatures fluctuated by up to 4°C. Doppler lidar observed vertical velocities collocated with the CopterSondes (Bell et al., 2020a, b) also indicate roughly 3 m s$^{-1}$ updrafts at the same time as the profiles in Figure 4. Turbulent transport of temperature therefore likely contributed to large spatial and temporal heterogeneity that can be detectable at the 10–20 m separation scales in this particular comparison flight.

Moreover, the horizontal winds throughout the depth of the atmosphere were weak and variable during these profiles, but observations from both aircraft demonstrate reasonable agreement (Figure 4c). As discussed previously, the CopterSonde estimates horizontal wind speeds and directions based on a second-order least-squares regression fit between the aircraft's tilt angle into the wind (calculated from three-dimensional Euler rotation matrices) and an Oklahoma Mesonet 10 m wind reference (Greene, 2018; Segales et al., 2020a). As more sophisticated autopilot-based adaptive wind estimation techniques become available, future studies should leverage this particular dataset along with other ground-based sensors (Bell et al., 2020b) or large eddy simulations (Pinto et al., 2020) to examine the effects of spatial and temporal heterogeneity on instruments located less than 100 m apart. For a more detailed perspective on the relative accuracy and precision of this dataset, readers are again referred to Barbieri et al. (2019) and Bell et al. (2020a).

### 4.3 Boundary Layer Transition – 18 July 2018

The morning of 18 July 2018, featured weak ambient synoptic-scale weather conditions and clear skies throughout the valley that enabled targeted measurements of the diurnal ABL transition. On this day, the two CASS teams were situated at the K04V and MOFF sites, with operations taking place from 1230–1945 UTC (0630–1345 MDT; Table 4). At both locations, vertical profiles to 914 m AGL were flown once every 15 min for the majority of the day. This particular case exemplified a canonical morning ABL transition, marked by a surface-based temperature inversion with a residual layer apparent atop beginning around 300 m at local sunrise, which occurred at 5:55 AM MDT (Figure 5, vertical red dashed line). After the sun rose above the Rocky Mountains and flooded the valley with shortwave radiation, vertical mixing dominated the lowest levels of the ABL. A surface-based dry adiabatic layer became present around 1415 UTC (Figure 5, red rectangle) as the ABL grew in depth. Surface-based vertical mixing appears to dominate the surface layer for several hours this morning, as the atmosphere above 300 m remains relatively steady-state for most of the early growth of the convective boundary layer. Entrainment-based heating of the growing ABL is also apparent by tracking the level of the strongest vertical potential temperature gradient through the morning (Figure 5, red oval).

### 4.4 Drainage Flow – 19 July 2018

The morning of 19 July 2018, was characterized by relatively quiescent conditions, making it the target of katabatic drainage flow observations. CASS flew CopterSondes from two locations (MOFF and NSF), conducting 46 total vertical profiles at 15 min intervals between 1115–1700 UTC (Table 4). A cursory glance at potential temperature in time-height coordinates from this day at the NSF site (Figure 6) reveals a similar ABL transition as observed on the previous day from the MOFF

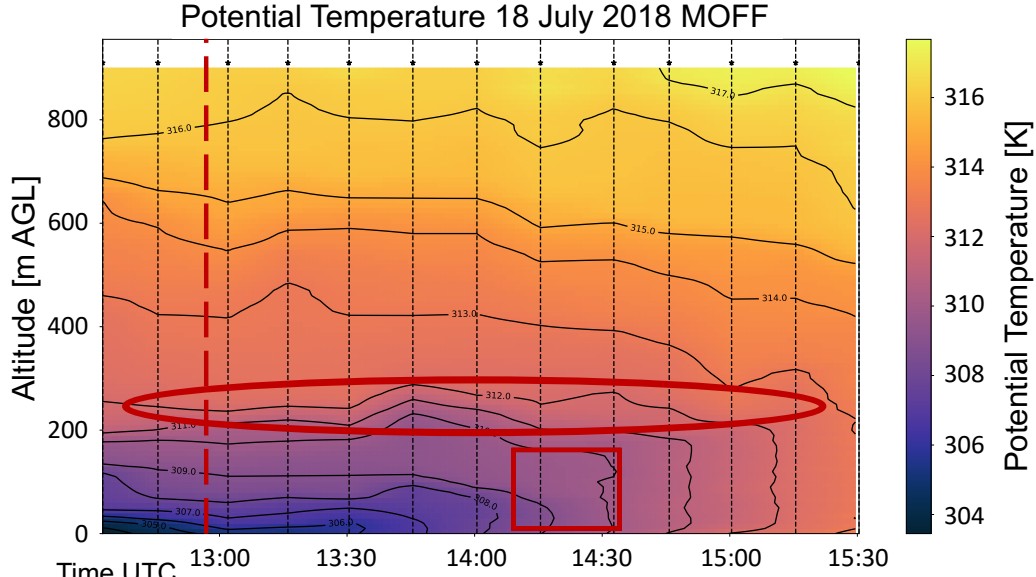

**Figure 5.** Time-height cross-section view of CopterSonde potential temperature observations at the MOFF on 18 July 2018. Shaded and contoured are potential temperature (K), and vertical dashed lines represent the time of CopterSonde flights. The star markers represent the maximum altitude achieved for a given profile. Values are interpolated through time at each level between successive flights. See text for discussion on the features annotated in red.

site (Figure 5). Wind speed and direction (Figure 7a, b) are also weak but primarily from the east below 400 m and from the north above 500 m. However, closer inspection reveals several harmonic structures, with a relative warm anomaly around 100 m AGL with a period of roughly 15 min at 1315 UTC propagating vertically through the next 4–5 profiles (Figure 6, red oval). Due to the highly stratified nature of the atmosphere at this time and the presence of complex topography, it is possible these disturbances are the result of internal gravity waves. Further investigation is necessary to elucidate specific details on this phenomenon and its sensitivity to spatial and temporal interpolation schemes.

## 5 Data availability

The OU CopterSonde data files from the LAPSE-RATE campaign are available for public access from the Zenodo data repository (Greene et al., 2020, DOI:10.5281/zenodo.3737087). They are in netCDF format with self-describing metadata and are organized by flight, aircraft tail number, and location. Included in each file are quality-controlled thermodynamic (temperature, pressure, humidity) and kinematic (wind speed and direction) measurements from collected by CASS during the LAPSE-RATE campaign from 14–19 July 2018. These files are from the fleet of 3 individual CopterSonde rotary-wing RPAS used during the campaign.

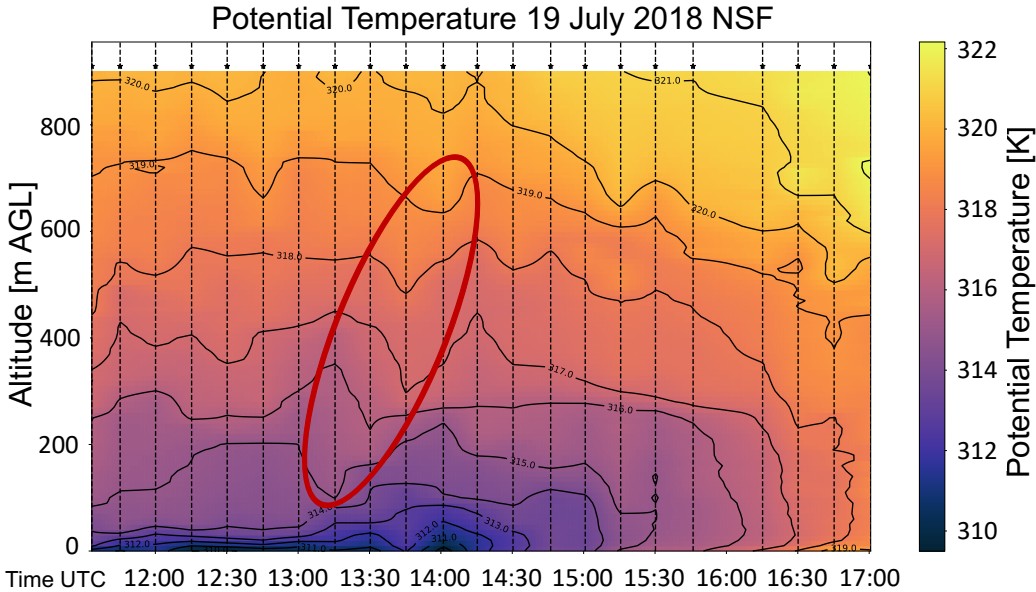

**Figure 6.** Time-height cross-section view of CopterSonde potential temperature observations at the North Star Farm (NSF) location on 19 July 2018. Figure notations follow the same conventions as in Figure 5. See text for discussion on the features annotated in red.

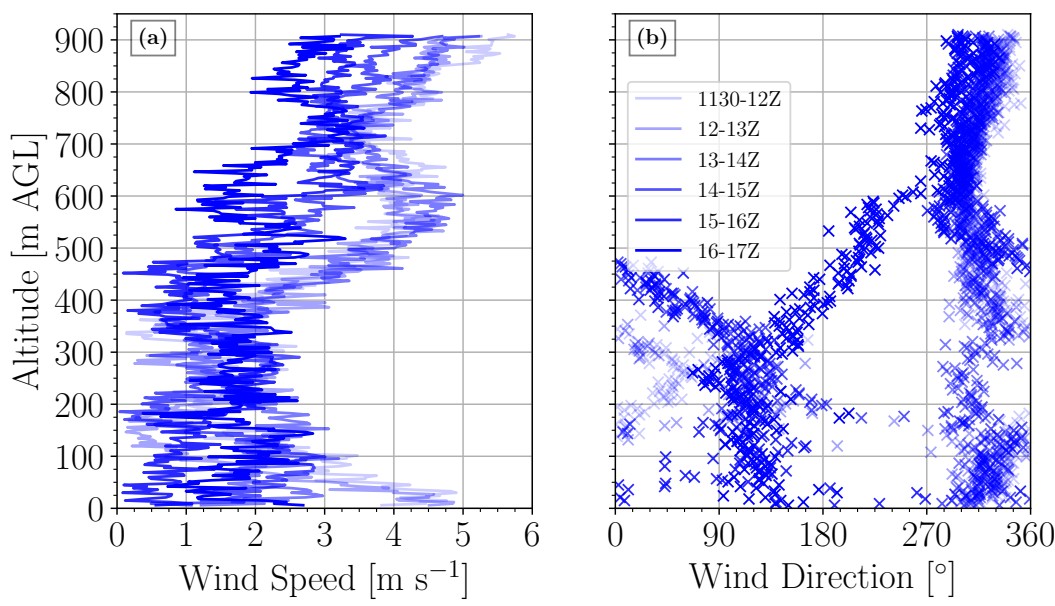

**Figure 7.** Hourly average profiles of wind speed (a) and wind direction (b) at the North Star Farm locations 19 July 2018.

The files follow a naming convention of UOK.ppppp.lv.yyyymmdd.hhmmss.cdf, where ppppp is a platform identification code (one of "OU944", "OU946", "OU955"), lv is the data file processing level, and yyyymmdd.hhmmss is the year-month-date.hour-minute-second of the file start date and time. Further information is included in the readme text file in this repository.

## 6  Summary

During July 2018, researchers from OU's CASS participated alongside federal and university partners in the LAPSE-RATE field campaign in San Luis Valley, Colorado, USA. The OU team successfully completed 180 flights using three RPAS over the course of six days of operation to collect vertical profiles of the thermodynamic and kinematic state of the ABL. This article describes sampling strategies, data collection, platform intercomparability, data quality and processing, and the dataset's possible applications to convective initiation, drainage flows, and ABL transitions.

The data available from these flights provides measurements of temperature, humidity, pressure, wind speed, and wind direction at a higher spatiotemporal resolution in the ABL than many conventional strategies, such as radiosondes, which will significantly contribute to characterizing the ABL within the San Luis Valley during the campaign. The data collected from the operations and platforms described here are uploaded and available for download through the Zenodo data repository (DOI:10.5281/zenodo.3737087). This data has already been featured in studies comparing RPAS profiling accuracies to those from more traditional profilers such as radiosondes, LIDARs, and AERIs (Bell et al., 2020a; Segales et al., 2020a). It will also be utilized in several ongoing studies that examine the use of RPAS data to improve temperature forecasts impacted by valley drainage flows (de Boer et al., 2020a) and the utility of using buoyancy as a metric to understand convection initiation and boundary layer transitions (Lappin and Chilson, 2021, in preparation).

*Author contributions.* Field Campaign Planning: E.P.L and P.C.; Data Collection: E.P.L, B.G., T.B., A.S., G.A., W.D, S.K., D.T., and P.C.; Data processing and quality control: B.G.; Data Analysis and Visualization: B.G., F.L., and T.B.; Writing: E.P.L, B.G., and F.L.; Supervision: P.C. and E.P.L; Funding acquisition: P.C.

*Competing interests.* The authors declare that they have no conflicts of interest.

*Acknowledgements.* This research has been supported in part by the National Science Foundation under Grant No. 1539070 and internal funding from the University of Oklahoma Office of the Vice President for Research and Partnerships. The authors would also like to thank the Moffat Consolidated School for the use of their parking lot and facilities for the duration of the campaign as well as the greater San Luis Valley community for welcoming the LAPSE-RATE campaign and providing land access for flight operations. Additional support for field operations in Colorado were provided by University of Oklahoma undergraduate research assistants Brandon Albert, Samantha Bashcky,
Christopher Hughes, Mark Thiel, and Brent Wolf.

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
