# Peer review of "Observations of the thermodynamic and kinematic state of the atmospheric boundary layer over the San Luis Valley, CO using the CopterSonde 2 remotely piloted aircraft system in support of the LAPSE-RATE field campaign"

_Earth System Science Data, 2020_

## Referee Comment (RC1) · Anonymous Referee #1 · 14 Sep 2020

**Title:** Observations of the thermodynamic and kinematic state of the atmospheric boundary layer over the San Luis Valley, CO using remotely piloted aircraft systems during the LAPSE-RATE field campaign

**Summary:** The manuscript describes sampling strategies and data collection using remotely piloted aircraft (RPA). Additionally, there are sections on platform inter-comparability, data quality, and processing. Lastly, techniques are described to evaluate the thermodynamic and kinematic state of the atmospheric boundary layer (ABL) over complex terrain with focus on applications for convective initiation, drainage flows, and ABL transitions.

**Recommendation:** The authors present the results from an interesting and unique field campaign. I recommend publication with minor revisions.

**Key points:**

I suggest reorganizing the manuscript a bit for clarity. Section 4 "Data Processing", comes at the end of the paper but it would strengthen the conclusions in Section 3 "Examples of Flight Data" if Section 4 was moved earlier into Section 2.1 "Description of the CopterSonde". Along this line of thinking I suggest moving Table 4 out of the summary section and showing it earlier in the paper. In the summary it is suggested to include larger implications to the data collection and analysis such as if the datasets collected throughout the six days led to improved forecasts for the San Luis Valley or did the campaign provide an avenue for increased use of RPAs in WMO, NOAA, or NCAR field campaigns? Line 19 of the introduction mentions, "unique opportunity to undertake an intensive comparison of the sensing capabilities of the aerial systems being utilized as a part of the campaign." But the summary does not reiterate the reason this opportunity was unique or its lasting implications.

It is nice to see the larger detail in figures 3 and 4 but it would help the reader in the discussion of comparisons if the figures were side by side or closer together.

Section 3.2 would be strengthened with more discussion on accuracy and precision of the dataset rather than just listing references so moving Section 4 earlier can address this. Additionally, adding in comparison data on figures from the radiosonde flights, CLAMPS AERI and Doppler LIDAR observations would be beneficial.

The following suggested changes are to help with clarity;

**Line 35-36**: Type of sensors (WMO approved)? Moving table 4 up would be helpful here.

**Line 44-45:** "Section 6 will provide concluding remarks about the dataset as well as future outlooks regarding the future applications of the dataset." The summary section does not seem to currently include "outlooks regarding the future applications of the dataset."

**Figure 1 and Line 56:** An immediate question for the reader is how the props influence the atmospheric sensors when viewing figure 1 then on line 56 it is mentioned the props were changed. Including a sentence or two on how prop wash has been considered would be helpful to the reader.

**Line 64-69:** Resolution of sensor measurements differ among variables. Moving lines 186 – 190 here would be helpful to the reader.

**Line 123 – 124:** Why different ascent and decent rates? Are rates optimized for sensor accuracy accounting for airflow? Was 10s loiter data kept? Did you use separate surface platform measurements to combine the last 10m of descent? Moving lines 199 – 208 here would be helpful.

**Figure 3:** The significant digits on the temperature contours seem to imply a measurement precision that is contradicted in table 4.

**Line 136 – 137:** It is mentioned that flight frequency changed between 15min and 30min for MOFF site but figures 3 and 4 both show changing flight frequency depending on time of day. It would be helpful to describe why flight frequency changed at particular times. For example, there is an hour between flights on figure 3 (1500 – 1600) and there is an increase in flight frequency on figure 4 from 1830 – 1944.

**Line 141:** "Figure 3 also shows the post convection cool down around 1800 UTC." This cool down is difficult to discern in the figure given the changing temperature contour separations and not knowing measurement precision (unless table 4 is moved earlier). It could help the reader to give actual temperature values or ranges to strengthen this observation.

**Line 154 – 155:** "While a small bias between the two aircraft exists in temperature . . ." At the surface and at 600m this looks to be almost 2 degrees which may not be small given the claim of a post convection cool down in figure 3. For all the graphs, does showing error bars make the graphs too difficult to read? Having the error bars could support the claim that the biases are small and winds show reasonable agreement.

**Line 157:** While it is helpful to have references on the accuracy and precision of the dataset, it is recommended the authors address this issue in at least a paragraph to support the claims of the inter-comparison flights similar to the explanations given in section 4 Data Processing.

**Line 165:** Please give the time for local sunrise.

**Line 230:** "intercompariblity" is misspelled. Intercomparability

---

## Referee Comment (RC2) · Anonymous Referee #2 · 22 Sep 2020

The authors have a great dataset illustrating phenomena in the boundary layer as measured by a RPAS. I would like to see a rewrite with more emphasis on that aspect as opposed to it reading like a data report see some examples below. . .

Line 20 This reads more like a data report or experimental field notes than an article on the uniqueness of RPAS for scientific discovery.

Line 55 Is it important to know that the carbon fiber blades were switched out??

[Figure]

Line 185 Why do we need to know that binary data was converted to JSON to CSV??

Figure 3 Temperature contours are plotted to the nearest .001 0C . I highly doubt that the authors have that kind of accuracy and if they do not the resolution of that parameter tied back to the accuracy or precision. Figure 5 would indicate that he precision is $\sim$ 1 0C. Table 4 would indicate +- 0.5 0C

[Figure]

---

## Referee Comment (RC3) · Anonymous Referee #3 · 23 Oct 2020

summary: The authors describe measurements of the CopterSonde 2 remotely piloted aircraft systems (RPAS) over complex terrain in the San Luis Valley, Colorado. The CopterSonde 2 and the flight strategy is briefly described, the data processing, availability and quality are discussed. The operations focused on convection initiation studies, diurnal transition studies, internal comparison flights and cold air drainage flows. Coordinated flights shell provide insight into the horizontal heterogeneity. The data set, as a part of the LAPSE-RATE campaign is publicly available.

[Figure]

general remarks: The introduction should explain the scientific goals of the LAPSE-RATE field campaign into more detail. Choice of location, previous measurements on the sight, typical and/or seasonal conditions, wind speeds and direction in this complex terrain with regard to synoptic conditions and so on. Further, the applied remote sensing techniques and the other measurement efforts during the campaign should be outlined in the introduction. A global overview of RPAS efforts for ABL studies should be given, rather than highlighting only OU's efforts in the field. The data processing chapter (4) should be moved to the description of the RPAS in Chapter 2 and the Data availability could be mentioned in Flight Strategies (2.2) alongside table 3, for example. The whole section 3 should be strengthened with more plots and details, comparisons to other measurement systems and further evaluations of the described atmospheric thermodynamic and kinematic state.

specific comments: L6 ff: The data from these coordinated flights provides insight into the horizontal heterogeneity of the atmospheric state over complex terrain as well as the expected horizontal footprint of RPAS profiles. What is meant with footprint? Footprint of the RPAS is confusing.

L18: What kind of conventional remote sensing techniques were applied?

L21: What are the scientific objectives?

L24-34: What about similar efforts of other institutions?

Figure 1: Does the manuscript include any data of that tower?

L64: Why is Table 4 in the very end and where are the accuracies coming from? What is meant by indirectly?

L64/L68/L69: Measurements at 10 and 20 Hz should be shown with a spectral analysis. Do the sensor resolve fluctuations that fast? Please provide spectra of an ascend of the copter to further discuss the resolution of the sensors.

L103-118: Is this section needed?

Figure 3 and Figure 4 should be next to each other

L137-143: Vague explanations. Please provide further details of how, where and when the feature of interest occur and why this implies the location of CI.

Section 3.2: The comparison should include other measurement systems like remote sensing devices, that were on sight. Further, the wind speed is too low in order to compare something. Both systems show unusual wind speed profiles, that do not agree. Maybe not much related to wind speed at all, but to attitude control parameters of the pixhawk autopilot system. Also the wind direction should be shown. Further comparison is needed, otherwise this section is not useful.

Section 3.3: Please provide further information. Time of sunrise and so on. L167 ff: Surface-based vertical mixing, above 300 m relatively steady-state for most of the early growth and entrainment-based heating of the growing ABL are only very briefly derived and need further

Figure 6 and Figure 7: It would be helpful to mark the features in the graphs and provide further data and graphs of the phenomena under discussion.

Section 3.4: Please provide further data and plots. What about wind speed and direction during this period?

L208: averaging intervals and time constants are fundamental. Why is it 1 s? Please provide further details and analysis.

L210: subjectively omitted? By hand? Algorithms should detect outliers systematically.

---

## Author Comment (AC1) · 3 Dec 2020

**The authors would like to thank Reviewer 1 for their comments on this paper. These comments have been reproduced here in black font color, and author responses are included in red.**

**Title:** Observations of the thermodynamic and kinematic state of the atmospheric boundary layer over the San Luis Valley, CO using remotely piloted aircraft systems during the LAPSE-RATE field campaign

**Summary:** The manuscript describes sampling strategies and data collection using remotely piloted aircraft (RPA). Additionally,there are sections on platform inter-comparability, data quality,and processing. Lastly, techniques are described to evaluate the thermodynamic and kinematic state of the atmospheric boundary layer (ABL) over complex terrain with focus on applications for convective initiation, drainage flows, and ABL transitions.

**Recommendation:** The authors present the results from an interesting and unique field campaign. I recommend publication with minor revisions.

**Key points**

I suggest reorganizing the manuscript a bit for clarity. Section 4 "Data Processing", comes at the end of the paper but it would strengthen the conclusions in Section 3 "Examples of Flight Data" if Section 4 was moved earlier into Section 2.1 "Description of the CopterSonde". Along this line of thinking I suggest moving Table 4 out of the summary section and showing it earlier in the paper. In the summary it is suggested to include larger implications to the data collection and analysis such as if the datasets collected throughout the six days led to improved forecasts for the San Luis Valley or did the campaign provide an avenue for increased use of RPAs in WMO, NOAA, or NCAR field campaigns? Line 19 of the introduction mentions, "unique opportunity to undertake an intensive comparison of the sensing capabilities of the aerial systems being utilized as a part of the campaign." But the summary does not reiterate the reason this opportunity was unique or its lasting implications.

We agree with the reorganization of the paper. Table 4 is now moved up to section 2 and is now Table 2. The Data Processing section will now come before the case study examples. We have added some text and references to highlight how this data is being used to the summary to solidify the projected impact of this data set. The last paragraph of the introduction has also been reworked to reflect all of the changes to the architecture of the paper.

As for the comment regarding line 19, we have added some text and references to the first introductory paragraph in lines 20-24. We hope this will more easily point readers to

information regarding the overall LAPSE-RATE campaign (de Boer et al BAMS 2020) as well as highlight a case study examining the utility of using RPAS data from LAPSE-RATE for forecasting applications (Glasheen et al JAIS 2020, Pinto et al ESSD 2020) and the ESSD paper highlighting the work forecasters did as a part of this effort.

It is nice to see the larger detail in figures 3 and 4 but it would help the reader in the discussion of comparisons if the figures were side by side or closer together.
Figures 3 and 4 were combined into one figure and the caption was changed to reflect the subfigures.

Section 3.2 would be strengthened with more discussion on accuracy and precision of the dataset rather than just listing references so moving Section 4 earlier can address this. Additionally, adding in comparison data on figures from the radiosonde flights, CLAMPS AERI and Doppler LIDAR observations would be beneficial.
Section 4 (data processing) was moved to now precede the data examples as Section 3. Since the AERI and LIDAR data are described in a separate paper, the appropriate citation was added and we chose not to overlay those data here for clarity.

The following suggested changes are to help with clarity;

Line 35-36: Type of sensors (WMO approved)? Moving table 4 up would be helpful here.
The data processing section now comes sooner after this, and references to tables 1 and 4 (now Table 2) were added to direct the reader to the appropriate information. An additional citation was added to Segales et al. (2020) which outlines this aircraft and the sensors used.

Line 44-45: "Section 6 will provide concluding remarks about the dataset as well as future outlooks regarding the future applications of the dataset." The summary section does not seem to currently include "outlooks regarding the future applications of the dataset."
Two sentences have been added starting at line 237 to highlight what studies this data set has already been utilized as a part of as well as upcoming papers that will be harnessing this data. We have also restructured this last paragraph of the introduction to point readers to efforts by other LAPSE-RATE teams as well as the overview and intercomparison efforts.

Figure 1 and Line 56: An immediate question for the reader is how the props influence the atmospheric sensors when viewing figure 1 then on line 56 it is mentioned the props were changed. Including a sentence or two on how prop wash has been considered would be helpful to the reader.

We added a comment that changing these propellers should not affect thermodynamic observations (lines 86-88) and also added a citation to Greene et al. (2019), which studied these effects on the same model of CopterSonde.

Line 64-69: Resolution of sensor measurements differ among variables. Moving lines 186 –190 here would be helpful to the reader.
We added the following to lines 79-80: "As will be discussed later, data from the different sensors were interpolated or downsampled so that all observations have a common time vector."

Line 123 –124: Why different ascent and decent rates? Are rates optimized for sensor accuracy accounting for airflow? Was 10s loiter data kept? Did you use separate surface platform measurements to combine the last 10m of descent? Moving lines 199 –208 here would be helpful.
We have added the following at lines 135-139: "As will be discussed in Section 3, only the ascent portion of these vertical profiles are considered for analysis. We therefore chose to fly slower on the ascent to maximize the vertical resolution when accounting for thermodynamic sensor response times. Moreover, by descending more rapidly we are able to achieve a higher maximum profile altitude than we would otherwise with the same battery configuration on the CopterSonde."

Figure 3: The significant digits on the temperature contours seem to imply a measurement precision that is contradicted in table 4.
This was an artifact of creating the figures in Python. The number of significant figures have been reduced to reflect the accuracy of the measurements.

Line 136 –137: It is mentioned that flight frequency changed between 15min and 30 min for MOFF site but figures 3 and 4 both show changing flight frequency depending on time of day. It would be helpful to describe why flight frequency changed at particular times. For example, there is an hour between flights on Figure 3 (1500 –1600) and there is an increase in flight frequency on Figure 4 from 1830 –1944.
The sentence was reworded to reflect when the flight frequency at MOFF changed and why. A sentence was added at ~line 195 was added to explain why the flight frequency increased after 1830 UTC at both sites.

Line 141: "Figure 3 also shows the post convection cool down around 1800 UTC." This cool down is difficult to discern in the figure given the changing temperature contour separations and not knowing measurement precision (unless table 4 is moved earlier). It could help the reader to give actual temperature values or ranges to strengthen this observation.

Two sentences were added (lines 200-204) were added to clarify the observed temperature difference. It will also help that the two figures are now next to each other.

Line 154 –155: "While a small bias between the two aircraft exists in temperature. . ." At the surface and at 600m this looks to be almost 2 degrees which may not be small given the claim of a post convection cool down in figure 3. For all the graphs, does showing error bars make the graphs too difficult to read? Having the error bars could support the claim that the biases are small and winds show reasonable agreement.

This is an interesting observation and we agree that additional context is warranted. The following was added in lines 231-241: In our experience with the CopterSonde, the discrepancies in the temperatures between the two identical platforms can be attributed to three main sources: 1) sunlight on an inadequately shielded sensor (discussed in Greene et al. 2019) at the correct relative angles of aircraft heading and sun zenith/azimuth; 2) natural variability in the atmosphere -- the 2 aircraft were 10-20 m apart, so this is not entirely unreasonable for a convective boundary layer; and/or 3) systemic bias related to calibration of the CopterSonde thermodynamic sensor package as a whole. While a combination of these three is the most likely explanation, we believe the spatial/temporal heterogeneity of the atmosphere during these observations should not be overlooked. For example, 3-second sonic anemometer temperatures from the Bailey et al. 2020 ESSD paper for this campaign reveal that during the 10-minute timeframe during these concurrent CopterSonde profiles (albeit at a different site but featuring similar land cover properties), 2-meter temperatures fluctuated by up to 4°C. Doppler lidar observed vertical velocities collocated with the CopterSondes (Bell et al. 2020a,b) also indicate ~3 m/s updrafts at the same time as the profile in this figure. Turbulent transport of temperature therefore likely contributed to large spatial and temporal heterogeneity that can be detectable at the 10 - 20 m separation scales in this particular comparison flight.

While further investigation into the relative contributions of these differences is beyond the scope of this paper, the context outlined above has been added for clarity. Here we also choose not to include error bounds, as the +/-0.5 °C accuracy from Table 4 (now Table 2) does not explicitly incorporate the effects of the spatial heterogeneity impacting the comparison of these two profiles that future studies may be interested in examining.

Line 157: While it is helpful to have references on the accuracy and precision of the dataset, it is recommended the authors address this issue in at least a paragraph to support the claims of the inter-comparison flights similar to the explanations given in section 4 Data Processing.

The "Data Processing" section has been moved forward to now be Section 3, so the following details have been added at the end of this section (lines 180-187): "In an effort to quantify the CopterSonde thermodynamic and kinematic observational biases relative

to a ubiquitous standard, Bell et al (2020a) compared vertical profile CopterSonde flights from LAPSE-RATE and in Oklahoma to collocated Vaisala RS92-SGP radiosondes. While unable to explicitly account for factors such as horizontal heterogeneity, the sample ranges in temperature, dewpoint temperature, and horizontal winds were large enough to determine baseline accuracies in each (Table 2). Namely, CopterSonde temperatures were within 0.5 °C of the radiosondes in the aggregate, which is largely due in part to the considerations taken for temperature sensor placement on-board the CopterSonde (Greene et al, 2018, 2019). Additionally, a broad intercomparison effort during the LAPSE-RATE campaign (Barbieri et al, 2019) resulted in similar statistics when comparing the CopterSonde observations to a common mobile meteorological reference."

Line 165: Please give the time for local sunrise.
Edited as suggested.

Line 230: "intercompariblity"is misspelled. Intercomparability
Edited as suggested.

---

## Author Comment (AC2) · 3 Dec 2020

**The authors would like to thank Reviewer 2 for their comments on this paper. These comments have been reproduced here in black font color, and author responses are included in red.**

The authors have a great dataset illustrating phenomena in the boundary layer as measured by a RPAS. I would like to see a rewrite with more emphasis on that aspect as opposed to it reading like a data report see some examples below.

Line 20 This reads more like a data report or experimental field notes than an article on the uniqueness of RPAS for scientific discovery.
Earth System Science Data is a journal that focuses on the dissemination of information about original datasets so that the data can be used by the broader scientific community, rather than the interpretation of these results. The authors believe that this article honors that spirit and was not meant to be an article on the novelty of using RPAS for scientific discovery. It very much is our field notes so that people can understand the strengths and limitations of our dataset.

Line 55 Is it important to know that the carbon fiber blades were switched out??
Yes, because it is an alteration of the platform's typical operating conditions that we have used or will use in other deployments of the CopterSonde RPAS. However, we have made the importance of this clearer based on this remark and feedback contained in the Open Discussion comments in lines 63-66.

Line 185 Why do we need to know that binary data was converted to JSON to CSV??
This information is relevant because the JSON format allows for the varying sampling rates for each data stream to coexist in the same file, whereas the conversion to CSV with a common time vector markedly simplifies reading and processing the data at this stage. This has been emphasized in the text in lines 149-154.

Figure 3 Temperature contours are plotted to the nearest .001 $^0$C. I highly doubt that the authors have that kind of accuracy and if they do not the resolution of that parameterized back to the accuracy or precision. Figure 5 would indicate that the precision is ~10C. Table 4 would indicate +- 0.5 $^0$C
For Figures 3 and 4, the contours appearing like they had an accuracy of 0.001 was actually an artifact of the plotting script. This has been corrected to reflect the correct accuracies as specified in Table 4 (now Table 2).

Figure 5 would indicate that the precision is $\square$ 1 0C. Table 4 would indicate +- 0.5 0C
Figure 5 (now Figure 4) has been updated so that the minor tick marks for temperature are every 0.5 °C to be closer to the posted accuracy in Table 4 (now Table 2).

---

## Author Comment (AC3) · 3 Dec 2020

**The authors would like to thank Reviewer 3 for their comments on this paper. These comments have been reproduced here in black font color, and author responses are included in red.**

summary: The authors describe measurements of the CopterSonde 2 remotely piloted aircraft systems (RPAS) over complex terrain in the San Luis Valley, Colorado. TheCopterSonde 2 and the flight strategy is briefly described, the data processing, availability and quality are discussed. The operations focused on convection initiation studies, diurnal transition studies, internal comparison flights and cold air drainage flows.Coordinated flights shell provide insight into the horizontal heterogeneity. The data set,as a part of the LAPSE-RATE campaign is publicly available.

general remarks: The introduction should explain the scientific goals of the LAPSE-RATE field campaign into more detail. Choice of location, previous measurements on the sight, typical and/or seasonal conditions, wind speeds and direction in this com-plex terrain with regard to synoptic conditions and so on. Further, the applied remote sensing techniques and the other measurement efforts during the campaign should be outlined in the introduction. A global overview of RPAS efforts for ABL studies should be given, rather than highlighting only OU's efforts in the field. The data processing chapter (4) should be moved to the description of the RPAS in Chapter 2 and the Data availability could be mentioned in Flight Strategies (2.2) alongside table 3, for example.The whole section 3 should be strengthened with more plots and details, comparisons to other measurement systems and further evaluations of the described atmospheric thermodynamic and kinematic state.

We have added additional clarification regarding the efforts and remote/in situ instrumentation that supplemented the RPAS data. We agree with the restructuring of the order of sections and have moved the data processing section to be Section 3. As this is an overview of OU's contribution to the campaign, we do not believe it is appropriate to provide a review of the state of the science in this article or detail the other institutions advancements. We have added additional citations to point to our collaborator's efforts in this campaign (lines 51-53) as well as direct interested parties to existing comprehensive reviews on the utilization of RPAS in weather and atmospheric science (lines 24-26).

We believe that providing additional plots and analysis is outside the scope of this article as it is meant to present the data set, discuss how it was collected, and how it can be utilized by other parties. Further analysis on these topics is forthcoming in Lappin et al 2021 and other planned publications.

specific comments:

L6 ff: The data from these coordinated flights provides insight into the horizontal heterogeneity of the atmospheric state over complex terrain as well as the expected horizontal footprint of RPAS profiles. What is meant with footprint? Footprint of the RPAS is confusing.

We agree that footprint is confusing here. We have deleted this phrasing from the sentence and have left the first half of the sentence to highlight how data from all teams could be utilized to highlight variations across the valley.

L18: What kind of conventional remote sensing techniques were applied?

Radiosondes, mobile mesonet units, CLAMPS, and LIDARs were all utilized as a part of the ground based in situ and remote sensing techniques that complimented the RPAS data collected by the participating institutions. That data will be presented in another publication in this special issue, Bell et al (2020b). The reference in the text has been clarified to include both remote and in situ sampling as well as point to this reference, around line 18.

L21: What are the scientific objectives?

As mentioned in lines 17-18 of the previous version, the objective of the campaign was to collect "targeted observations of cold air drainage flows, convection initiation, and morning boundary layer transitions" with RPAS, in situ, and remote sensing instruments. We have rephrased lines 14-20 and added to lines 38-40 to further clarify the campaign's objectives and how we contributed to them.

We have also added additional citations to direct readers to the campaign overview papers in the special issue and the Bulletin of the American Meteorological Society in lines 50-51.

L24-34: What about similar efforts of other institutions?

As this work focuses on OU's efforts to the campaigns, we will not be discussing our partner institutions here. However, we have added references to these teams' efforts that are also presented in this special issue in starting at line 51 to assist readers in finding this material.

Figure 1: Does the manuscript include any data of that tower?

No, because this is a photo taken back at our field laboratory in Washington, OK to showcase the RPAS. The data was collected in CO.

L64: Why is Table 4 in the very end and where are the accuracies coming from? What is meant by indirectly?

Table 4 has been brought to Section 2.1 and is now Table 2. The original thinking was that this table is the culmination of the processing and shows explicitly what users will find in the data files, but we acknowledge that this information is useful much more early on in the manuscript. The accuracies originate from the Bell et al. (2020a) study cited in the caption as compared to Vaisala RS92-SGP radiosondes. The "indirectly" comment has been removed for clarity.

L64/L68/L69: Measurements at 10 and 20 Hz should be shown with a spectral analysis. Do the sensor resolve fluctuations that fast? Please provide spectra of an ascend ofthe copter to further discuss the resolution of the sensors.
As described in Section 3, the thermodynamic and kinematic observations are averaged to 3 m altitude bins, which effectively removes the spectral information at the original sampled frequencies. Spectral analysis of these sensors is therefore out of the scope of this data overview paper.

L103-118: Is this section needed?
Yes - it is important to outline how one gets authorization to operate in the National Airspace for people wishing to conduct RPAS work in the future. This is a very important part of collecting the data and may not be obvious to individuals wishing to work with RPAS in the future.

Figure 3 and Figure 4 should be next to each other
Figure 3 and 4 have been combined into one figure.

L137-143: Vague explanations. Please provide further details of how, where and when the feature of interest occur and why this implies the location of CI.
This topic is further investigated in an Atmospheric Measurement Techniques paper, currently in preparation. Commentary about the motivations and methods in the upcoming paper were added in lines 206-209.

Section 3.2: The comparison should include other measurement systems like remote sensing devices, that were on sight. Further, the wind speed is too low in order to compare something. Both systems show unusual wind speed profiles, that do not agree. Maybe not much related to wind speed at all, but to attitude control parameters of the pixhawk autopilot system. Also the wind direction should be shown. Furthercomparison is needed, otherwise this section is not useful.
Because this paper specifically discusses the CopterSonde data collected during LAPSE-RATE, we intentionally chose not to include comparisons to other instruments; however, we have added citations to accompanying datasets in this section. As for the wind profiles, we agree that the comparison presented is not a perfect agreement. We have added a

profile of wind direction to Figure 4d. While deeper discussion into the mechanisms behind the possible disagreement is beyond the scope of this data paper, the following context has been added about how winds are derived (lines 234-241): "As discussed previously, the CopterSonde estimates horizontal wind speeds and directions based on a second-order least-squares regression fit between the aircraft's tilt angle into the wind (calculated from three-dimensional Euler rotation matrices) and an Oklahoma Mesonet 10 m wind reference (Greene et al. 2018, Segales et al. 2020). As more sophisticated autopilot-based adaptive wind estimation techniques become available, future studies should leverage this particular dataset along with other ground-based sensors (Bell et al. 2020b) or large eddy simulations (Pinto et al. 2020) to examine the effects of spatial and temporal heterogeneity on instruments located less than 100 m apart."

Section 3.3: Please provide further information. Time of sunrise and so on.
Local sunrise time in MDT has been added as suggested.

L167 ff:Surface-based vertical mixing, above 300 m relatively steady-state for most of the early growth and entrainment-based heating of the growing ABL are only very briefly derived and need further
Further discussion and analysis is beyond the scope of this paper, whose primary purpose is just to demonstrate the type of data included in this dataset.

Figure 6 and Figure 7: It would be helpful to mark the features in the graphs and provide further data and graphs of the phenomena under discussion.
These figures have been updated with larger font size and annotations.

Section 3.4: Please provide further data and plots. What about wind speed and direction during this period?
We have added a figure summarizing the wind speed and direction profiles during this timeframe. This is now Figure 7.

L208: averaging intervals and time constants are fundamental. Why is it 1 s? Please provide further details and analysis.
The following details have been added (lines 173-179): "Finally, the 3 m averaging interval was chosen under consideration of the average ascent rate (3 m/s) and an approximate time constant of the sensor payload of 2 s. This time constant is based upon experiments during the ISOBAR18 campaign with an older version of the CopterSonde and identical sensors (Kral et al., 2020; Greene et al., 2021, in preparation) where the aircraft was subjected to a series of quasi-step-function inputs between a sauna and the below-freezing environment of Hailuoto, Finland. The averaging interval of 3 m is therefore

approximately double the vertical resolution as predicted by the response time and ascent rate, so further studies will be needed to elucidate the impacts of these decisions."

L210: subjectively omitted? By hand? Algorithms should detect outliers systematically
We agree, and this is an ongoing effort to automate an objective process. With only 3 T and 3 RH sensors and no true "reference" for each vertical profile aside from a ground station (only occasionally), it is not always possible to determine a "most correct" sensor based just on simple statistics like mean and standard deviation. Therefore, our current method requires subjective inspection of each profile to determine which sensors perform the most similarly (i.e., highly correlate together). Usually there is high correlation and low spread, but occasionally the sensors strongly correlate but are separated by a large offset; other times, sensors weakly correlate but have a small offset. Since we have thus far been unable to determine objective thresholds for these features, a subjective perspective is required. This is the same technique in data processing for the vertical profiles compared in Bell et al. 2020a, which identified accuracies of +/-0.5 °C in temperature and +/-2% RH when compared to Vaisala RS92-SGP radiosondes often regarded as a "gold standard". We are therefore confident in this approach, although we do agree that more explanation is warranted.

Lines 180-185 now read as follows: "Because the CopterSondes were outfitted with 3 temperature and 3 RH sensors each, it was necessary to inspect each of their time-series outputs with respect to one another to determine potential outliers. Although an objective method of doing so is ideal, research into this is still ongoing and thus we chose to subjectively analyze each sensor individually. A given sensor was omitted from further consideration if it did not correlate with the other sensors and/or there was a large bias between them (greater than 0.5 °C)."